# Ecological Factors Associated with the Distribution of *Bemisia tabaci* Cryptic Species and Their Facultative Endosymbionts

**DOI:** 10.3390/insects14030252

**Published:** 2023-03-02

**Authors:** Hongran Li, Zhihui Jiang, Jincheng Zhou, Xin Liu, Youjun Zhang, Dong Chu

**Affiliations:** 1Shandong Engineering Research Center for Environment-Friendly Agricultural Pest Management, College of Plant Health and Medicine, Qingdao Agricultural University, Qingdao 266109, China; 2Shenzhen Branch, Guangdong Laboratory of Lingnan Modern Agriculture, Key Laboratory of Gene Editing Technologies (Hainan), Ministry of Agriculture and Rural Affairs, Agricultural Genomics Institute at Shenzhen, Chinese Academy of Agricultural Sciences, Shenzhen 515100, China; 3Department of Entomology, College of Plant Protection, Shenyang Agricultural University, Shenyang 110866, China; 4State Key Laboratory of Crop Stress Adaptation and Improvement, School of Life Sciences, College of Agriculture, Henan University, Kaifeng 475004, China; 5Department of Plant Protection, Institute of Vegetables and Flowers, Chinese Academy of Agricultural Sciences, Beijing 100081, China

**Keywords:** *Bemisia tabaci*, ecological niche, facultative endosymbionts, native cryptic species, multiple infection

## Abstract

**Simple Summary:**

*Bemisia tabaci* is a globally notorious agricultural pest due to its highly polyphagous characteristic, its composition as a cryptic species, and its transmission of plant viruses. The whitefly has endosymbiotic bacteria in cells inside their bodies that have strong effects on the host biology. Generally, the geographical distribution of these cryptic species and their endosymbionts’ infection patterns are related to ecological factors. In this study, we determined eight cryptic species (MED, MEAM1, Asia I, Asia II 1, Asia II 2, Asia II 6, China 1, and China 6) from 29 geographical localities across China. Native cryptic species were still widespread in the Yangtze River Valley and eastern coastal areas. In addition, we found that the cryptic species distribution and their facultative endosymbionts’ infection patterns were closely associated with the geographical and environmental factors, i.e., latitude and annual mean temperature. This work highlights that range expansion of invasive species and host–endosymbiont interactions need to be studied within an environmental and geographic context.

**Abstract:**

The sweetpotato whitefly, *Bemisia tabaci* species complex, comprises at least 44 morphologically indistinguishable cryptic species, whose endosymbiont infection patterns often varied at the spatial and temporal dimension. However, the effects of ecological factors (e.g., climatic or geographical factors) on the distribution of whitefly and the infection frequencies of their endosymbionts have not been fully elucidated. We, here, analyzed the associations between ecological factors and the distribution of whitefly and their three facultative endosymbionts (Candidatus *Cardinium hertigii*, Candidatus *Hamiltonella defensa*, and *Rickettsia* sp.) by screening 665 individuals collected from 29 geographical localities across China. The study identified eight *B. tabaci* species via mitochondrial cytochrome oxidase I (mt*COI*) gene sequence alignment: two invasive species, MED (66.9%) and MEAM1 (12.2%), and six native cryptic species (20.9%), which differed in distribution patterns, ecological niches, and high suitability areas. The infection frequencies of the three endosymbionts in different cryptic species were distinct and multiple infections were relatively common in *B. tabaci* MED populations. Furthermore, the annual mean temperature positively affected *Cardinium* sp. and *Rickettsia* sp. infection frequencies in *B. tabaci* MED but negatively affected the quantitative distribution of *B. tabaci* MED, which indicates that *Cardinium* sp. and *Rickettsia* sp. maybe play a crucial role in the thermotolerance of *B. tabaci* MED, although the host whitefly per se exhibits no resistance to high temperature. Our findings revealed the complex effects of ecological factors on the expansion of the invasive whitefly.

## 1. Introduction

Biological invasions can lead to serious negative effects on the indigenous ecosystem, biodiversity, and economy [1,2,3]. Understanding the ecological niche is a crucial factor for demonstrating distribution patterns and developing prevention and control strategies for invasive species [4]. During colonization in new habitats, invasive species usually compete with native species for sharing the same niche space [5]. However, in most cases, when introduced to new ranges, invasive species often face novel biotic interactions and dispersal limitations, resulting in recurring extinction events and limited dispersal ability, which can have serious implications for full colonization [6,7]. Furthermore, colonization can also be limited by various abiotic factors, such as geographical barriers and/or non-climatic environmental factors [8]. Thus, ecological factors’ association with the population expansion of exotic species must be revealed, as this will help us understand the phenomenon of successful invasion of exotic species. 

Moreover, invasive species often form a mutualistic relationship with their symbionts in the introduced regions, which particularly enhance the potential adaptations after invasion [9,10,11,12]. Although symbionts can play essential roles in the adaptation of their host insects, the infection dynamics in the same host populations can be influenced on ecological and geographic factors, such as temperature, precipitation, vegetation, longitude, latitude, and altitude [13,14,15,16]. For example, the diversity and infection frequencies of endosymbionts are directly impacted by climatic and ecological factors in chestnut weevil, *Curculio sikkimensis* [17]. The infection pattern of endosymbionts in spider mites are closely associated with environmental factors, i.e., higher annual mean temperatures increased *Wolbachia* sp. infection frequency, while higher altitudes increased *Spiroplasma* sp. and Candidatus *Cardinium hertigii* infection frequency [16]. In pea aphid *Acyrthosiphon pisum*, the geographic distribution of endosymbionts in natural populations shows a strong association with host plant species, precipitation, and temperature [13]. These findings demonstrate that the natural environment and endosymbiont infection frequencies are closely associated. These associations could be the result of the direct or indirect impact of environmental factors on endosymbionts and/or their hosts [18]. 

The sweetpotato whitefly, *Bemisia tabaci* (Gennadius) species complex, is a notorious agricultural pest worldwide comprising at least 44 cryptic species that differ in many biological traits, such as virus transmission and insecticide resistance [19,20]. Among these cryptic species, Middle East-Asia Minor 1 (MEAM1) and Mediterranean (MED) have been most invasive and destructive [21,22]. In China, MEAM1 was first introduced in the middle to late 1990s, which then displaced the native cryptic species (NCS) by its strong displacement capacity [23,24]. Since 2008, MED rather than MEAM1 has been the dominant cryptic species in the field in most regions of China [25]. Although MEAM1 and MED have been widespread in the field, NCS still existed in some restricted locations, probably due to the geographical barrier [24,26,27]. Facultative symbionts, including Candidatus *Cardinium hertigii*, Candidatus *Fritschea*, Candidatus *Hamiltonella defensa*, Candidatus *Hemipteriphilus asiaticus*, *Rickettsia* sp., *Arsenophonous* sp., and *Wolbachia* sp., were found among members of the *B. tabaci* complex [28,29,30], and some of them may contribute directly to host fitness benefits and manipulate host reproduction and sex ratio. For example, *Rickettsia* sp. dramatically increases fitness and alters sex ratios under laboratory conditions for *B. tabaci* MEAM1 [10], while *Cardinium* sp. conferred the fitness cost to *B. tabaci* MED [31]. The infection pattern of facultative endosymbionts has been distinct. For example, *Hamiltonella* sp. infection has been reported only in *B. tabaci* MED and MEAM1 [29,32]. In addition, the endosymbiont prevalence varies across different cryptic species. For example, the infection frequency of *Wolbachia* sp. was significantly higher in NCS than in MED or MEAM1 [33,34]. However, the effects of ecological factors (e.g., climatic or geographical factors) on the distribution of *B. tabaci* species, particularly the infection frequency of their endosymbionts, have not been well studied.

Here, to determine the effects of ecological factors on the distribution pattern of *B. tabaci* species and the infection frequency of their endosymbionts, we analyzed the composition of *B. tabaci* cryptic species and three common endosymbiont lineages, namely *Cardinium* sp., *Hamiltonella* sp., and *Rickettsia* sp., by screening 665 individuals collected from 29 geographical locations across China in 2021. The aims of our study were to (i) investigate the suitable habitat distribution of *B. tabaci* cryptic species, (ii) examine the endosymbiont distribution and the infection patterns in different cryptic species, and (iii) determine whether and how the infection patterns are related to the local climate.

## 2. Materials and Methods

### 2.1. Whitefly Sampling and DNA Extraction

A total of 41 *B. tabaci* populations were collected from nine host plants (abutilon avicennae, cotton, cucumber, eggplant, pepper, pumpkin, soybean, tobacco, and tomato) in 29 geographic locations across China in 2021 (Appendix A). At least 15 individuals per locality were sampled for whitefly species identification. After removing the *B. after* and *B. subdecipiens* individuals in some of the collections, a total of 655 *B. tabaci* adults were prepared for the further symbiont detection and correlation analysis. Details of sampling are summarized in Appendix A. All live adults were placed in tubes containing 95% alcohol and stored at −20 °C prior to DNA extraction. Total genomic DNA was extracted from individual *B. tabaci* insects by grinding them in a PCR tube containing 30 µL of lysis buffer (500 mmol/L Tris, 1.0% SDS, 1.0 mmol/L EDTA, 20 mmol/L NaCl, and 200 mg/mL protease K). Lysis buffer was prepared according to a standard protocol [35]. The homogenate was transferred to a 0.2 mL micro tube and incubated in a thermocycler at 65 °C for 15 min and 95 °C for 10 min. The processed samples were stored at −20 °C.

### 2.2. PCR Amplification

A total of 655 individuals were prepared for cryptic species identification and endosymbiont screening. The primer of mt*CO1* gene was used for whitefly cryptic species identification. The 16S rDNA primers were used to detect Candidatus *Portiera aleyrodidarum*, *Wolbachia* sp., *Rickettsia* sp. *Hamiltonella* sp., and *Cardinium* sp., and 23S rDNA primers were utilized for *Arsenophonus* sp. and *Fritschea* sp. All the specific primers, gene features, and annealing temperatures are listed in Appendix A. The PCR reaction mix consisted of 22 μL of 1.1X buffer (Tsingke Biotech, Qingdao, China), 1 μL of 10 μM of each primer, and 2 μL template DNA. Cycling conditions consisted of initial denaturation at 98 °C for 5 min, followed by 35 cycles of 94 °C for 45 s, annealing for 1 min, at 72 °C for 1.5 min, and final extension at 72 °C for 10 min. The PCR products were electrophoresed with one negative control (sterile water instead of DNA) and positive controls (DNA from previous sequencing) on an ethidium-bromide-stained 1.0% agarose gel. All amplified products were directly sequenced using an ABI 3730 DNA analyzer (Qingdao, China).

### 2.3. Sequence Alignment and Phylogenetic Analysis

All sequences obtained from the current study were edited and aligned using Clustal W in MEGA v6.0 software [36] and the haplotypes were determined using DnaSP v5.032 [37]. The haplotypes were defined when at least one mutational site occurred between two sequences [38]. To determine the composition of cryptic species in the present study, we firstly retrieved the reference sequences representing 43 known *B. tabaci* cryptic species (including native and invasive cryptic species) from NCBI database following the previous report by Kanakala and Ghanim, 2019. Then, using 17 haplotype sequences obtained in this study and 43 retrieved reference sequences, we constructed the neighbor-joining (NJ) and maximum likelihood (ML) phylogenetic trees in MEGA v6.0 software [39,40]. The Kimura 2-parameter model was applied due to the sites with gaps (insertions and/or deletions) being removed in all aligned sequences as the model does not take into account the evolution by insertions and deletions [41]. Bootstrap values were based on 1000 replicates and *B. afer*, *B. atriplex*, *B. subdecipiens*, and *Trialeurodes vaparariorum* were used as outgroups. Finally, the cryptic species were determined when the bootstrap value was >0.99 between haplotype sequences and retrieved reference sequences.

### 2.4. Statistical Analyses

Owing to the small population size of six indigenous cryptic species, we merged them as the NCS group to conduct statistical analyses. The random forest model was used to predict the distribution of MED, MEAM1, and NCS and to determine the effects of the annual mean temperature (AMT), annual precipitation (AP), latitude, and longitude on their distribution [42]. The structural equation model (SEM) with the Satorra–Bentler correction was used to estimate the effects of geographical and climatic factors on the distribution of *B. tabaci* MED, MEAM1, and NCS [43]. To avoid errors induced by deviance of the parameters, a standardized coefficient was adopted to estimate the linear relationship of every model path. We finally selected the simplest model having the lowest Akaike’s information criterion (AIC) value.

Although we detected the seven frequent endosymbionts in *B. tabaci* cryptic species, the infection frequency of *Wolbachia* sp., *Arsenophonus* sp., and *Fritschea* sp. were too low to conduct the further correlation analysis. Thus, the infection frequencies of *Cardinium* sp., *Hamiltonella* sp., and *Rickettsia* sp. in *B. tabaci* MED, MEAM1, and NCS were analyzed using one-way analysis of variance (ANOVA) with binomial distribution. The Tukey–Kramer test was applied for post hoc multiple comparison of the model. Given the low infection frequency of the three endosymbionts in *B. tabaci* MEAM1 and NCS, only the effects of geographic and climatic factors on endosymbiont infection frequency in *B. tabaci* MED were further analyzed. In addition, sampling sites with less than nine MED individuals were excluded from the correlation analysis. Climatic data for each geographical site were derived from the online WORLDCLIM database (https://www.worldclim.org//, accessed on 1 October 2022) [44]. To estimate the effects of geographic and climatic factors on the infection frequencies of *Cardinium* sp., *Hamiltonella* sp., and *Rickettsia* sp. in MED, the SEM with Satorra–Bentler correction was also adopted.

To investigate whether different infections have a tendency to co-occur in different populations, the expected frequency of endosymbionts’ coinfection was computed on the basis of the observed endosymbiont infection frequencies in each sample. Then, the binomial sign test was used to test the significant difference in the double or triple infection frequency between the observed and expected values. Due to low coinfection in MEAM1 and NCS, this analysis could only be conducted in MED samples for the *Hamiltonella* sp.–*Rickettsia* sp. combination, *Cardinium* sp.–*Rickettsia* sp. combination, *Cardinium* sp.–*Hamiltonella* sp. combination.

Except where explicitly stated otherwise, all statistical analyses and data processing were conducted in R v4.0.1 (R 2016).

## 3. Results

### 3.1. Identification and Distribution of B. tabaci Cryptic Species

A total of 17 haplotypes were identified from the 665 whitefly individuals collected in 2021. The selected individuals comprised two invasive species and six NCS (Asia I, Asia II 1, Asia II 2, Asia II 6, China 1, and China 6) (Figure 1; Appendix A). Among them, MED was the dominant species (66.9%), followed by the six NCS (20.9%) and MEAM1 (12.2%).

The field investigation showed that *B. tabaci* MED and MEAM1 were widely distributed in northern and southern China, respectively, while the NCS were prevalent in the eastern coastal areas (Figure 2). *B. tabaci* MED was predominantly distributed in Beijing, Liaoning, Shandong, Henan, Jiangsu, and Guangdong provinces at the ratios of 100.0%, 73.0%, 95.4%, 92.3%, 76.5%, and 100.0%, respectively. *B. tabaci* MEAM1 was predominantly distributed in Hainan Province at a ratio of 70.7%. Among the six NCS, China 1 was the most widespread and was present in the 15 provinces surveyed (average ratio: 12.6%). Asia II 1 was the second most dominant indigenous species identified in the present investigation and was only found in Zhejiang and Hainan provinces as the second most dominant NCS in the present investigation (average ratio: 3.6%). The other NCS including Asia I, Asia II 2, and Asia II 6 occurred occasionally, with a percentage of <3.6% in the field (Figure 2).

### 3.2. Habitat Distribution of B. tabaci Cryptic Species in China

The random forest model showed the effects of different factors, namely AMT, AP, latitude, and longitude, on the frequency of *B. tabaci* cryptic species. The effect of these factors on the frequency of MED, MEAM1, and NCS was in the orders of latitude > AP > AMT > longitude; AMT > latitude > longitude > AP; and latitude > AMT > AP > longitude, respectively (Figure 3A). Moreover, the distribution range of the invasive species was generally larger than that of NCS. Among these species, the two invasive species exhibited different habitat distributions. The NCS was mainly distributed in southern and central China, while MEAM1 was mainly spread over southern China (Figure 3B).

### 3.3. Geographic Distribution of Endosymbionts in B. tabaci Cryptic Species

In MED populations, the infection frequencies of *Cardinium* sp., *Hamiltonella* sp., and *Rickettsia* sp. were variable among different sampling locations (Figure 4). *Hamiltonella* sp. and *Rickettsia* sp. exhibited relatively high frequencies in all collections, while *Cardinium* was sporadically distributed in sampling locations, exhibiting relatively low frequencies in northern sampling locations and high frequencies in southern sampling locations. In MEAM1 populations, no one was infected by *Cardinium* sp. (Appendix A). In NCS populations, the distribution titer of the three endosymbionts was low, e.g., *Cardinium* sp. was only detected in Asia II 2, *Hamiltonella* sp. was detected in China 1 and Asia I, and *Rickettsia* sp. was present in Asia I, Asia II 1, Asia II 2, and China 1 (Appendix A). Because the number of individuals within the population was lower in *B. tabaci* MEAM1 and NCS, the downstream analysis was performed only for MED populations.

### 3.4. Infection Frequencies of Facultative Endosymbionts in B. tabaci Cryptic Species

The infection frequency of *Cardinium* sp. was 13.0% (56/430) in *B. tabaci* MED and 1.4% (2/139) in NCS, whereas this endosymbiont was not found in any of the specimens of *B. tabaci* MEAM1 (Figure 4). The *Cardinium* sp. infection frequency was significantly affected by cryptic species (χ^2^ = 25.67, df = 2, *p* < 0.001), that is, the frequency of *Cardinium* sp. was significantly higher in MED than that in MEAM1 (z = 2.40, *p* = 0.043) and NCS (z = 9.38, *p* = 0.0060) (Figure 5A).

The infection frequency of *Rickettsia* sp. was 79.3% (341/430) in MED, 39.5% (32/81) in MEAM1, and 10.8% (15/139) in NCS and was significantly influenced by cryptic species (χ^2^ = 203.05, df = 2, *p* < 0.001); the infection frequency of *Rickettsia* sp. was significantly higher in MED than in MEAM1 (z = 6.28, *p* < 0.001) and NCS (z = 10.93, *p* < 0.001). Additionally, the frequency of *Rickettsia* was significantly higher in MEAM1 than in NCS (z = 4.66, *p* < 0.001) (Figure 5B).

The infection frequency of *Hamiltonella* sp. was 92.3% (397/430) in *B. tabaci* MED, 71.6% (58/81) in *B. tabaci* MEAM1, and 3.6% (5/139) in *B. tabaci* NCS. The infection frequency of *Hamiltonella* sp. was significantly affected by *B. tabaci* cryptic species (χ^2^ = 124.86, df = 2, *p* < 0.001). The infection frequency of *Hamiltonella* sp. was significantly higher in MED than in MEAM1 (z = 5.51, *p* < 0.001) and NCS (z = 8.07, *p* < 0.001) (Figure 5C).

### 3.5. Multiple Infections of Facultative Endosymbionts in B. tabaci MED

The coinfection frequencies of the three endosymbionts in *B. tabaci* MED were calculated and further analyzed. A Wilcoxon paired test demonstrated that the coinfection frequency of *Cardinium* sp.–*Hamiltonella* sp.–*Rickettsia* sp. was significantly lower than that expected across populations (eight populations, *p* < 0.05), whereas the other coinfections had no significant difference with expected frequency, such as *Cardinium* sp.–*Hamiltonella* sp. (*n* = 9, *p* = 0.31), *Cardinium* sp.–*Rickettsia* sp. (*n* = 8, *p* = 0.056), or *Hamiltonella* sp.–*Rickettsia* sp. (*n* = 16, *p* = 0.46) (Figure 6, Table 1).

### 3.6. Association of Environmental Factors on the Frequency of Cryptic Species and Facultative Endosymbionts in MED

The structural equation model (SEM) was accepted after excluding eight paths (χ^2^ = 3.67, df = 20, *p* = 0.45, comparative fit index (CFI) = 1.000, AIC = 44.22, SRMR = 0.023) (Figure 7). The results showed that AMT significantly decreased with latitude (−0.95 ± 0.023, z = 41.56, *p* < 0.001) and AP significantly increased with longitude (0.45 ± 0.052, z = 5.63, *p* < 0.001) but decreased with latitude (−0.99 ± 0.043, z = 25.15, *p* < 0.001). The frequency of *B. tabaci* MED significantly decreased with AMT (−0.33 ± 0.20, z = 2.67, *p* = 0.0093). The frequency of *B. tabaci* MEAM1 significantly increased with longitude (0.41 ± 0.15, z = 2.82, *p* = 0.0051) and AMT (0.60 ± 0.27, z = 2.23, *p* = 0.026) but decreased with AP (−0.57 ± 0.28, z = 2.054, *p* = 0.040). The frequency of NCS significantly increased with latitude (0.60 ± 0.28, z = 2.11, *p* = 0.035) and AP (0.75 ± 0.30, z = 2.51, *p* = 0.012) but decreased with longitude (−0.22 ± 0.099, z = 2.25, *p* = 0.024). In addition, when the frequency of NCS was low, the higher frequencies in MED (−0.88 ± 0.044, z = 20.16, *p* < 0.001) and MEAM1 (−0.46 ± 0.13, z = 3.48, *p* < 0.001) were observed. However, the correlation between MED and MEAM1 was nonsignificant (−0.013 ± 0.13, z = 0.093, *p* = 0.93) (Figure 7A). Overall, the frequency of *B. tabaci* cryptic species may be driven by geographic factors (i.e., latitude and longitude) and climatic factors (i.e., AMT and AP) (Table 2).

The infection frequency of *Cardinium* sp. significantly increased with AMT (0.43 ± 0.13, z = 3.39, *p* < 0.001), while the frequency of *Rickettsia* sp. significantly increased with latitude (0.99 ± 0.15, z = 6.76, *p* < 0.001) and AMT (0.88 ± 0.13, z = 6.71, *p* < 0.001). The frequency of *Hamiltonella* sp. significantly increased with latitude (0.61 ± 0.11, z = 4.32, *p* < 0.001) but decreased with longitude (−0.17 ± 0.041, z = 4.16, *p* < 0.001). Meanwhile, the frequency of *Cardinium* sp. negatively correlated with that of *Rickettsia* sp. (−0.17 ± 0.041, z = 4.16, *p* < 0.001); however, the correlations were nonsignificant between *Cardinium* sp. and *Hamiltonella* sp. and between *Rickettsia* sp. and *Hamiltonella* sp. (Figure 7B). Overall, the infection frequencies of the three endosymbionts may be driven by geographic factors (i.e., latitude, longitude, and altitude) and climatic factors (i.e., AMT and AP) (Table 3).

## 4. Discussion

Here, we conducted an extensive field survey to investigate the distribution of whitefly complex species and the infection pattern of their bacterial endosymbionts in natural populations across China (Appendix A). Eight complex species were identified via mt*COI* sequence alignment, including two invasive species (MED and MEAM1) and six NCS (Asia I, Asia II 1, Asia II 2, Asia II 6, China 1, and China 6). Previous surveys have indicated that, except for the MED and MEAM1, a total of 18 NCS were occurred in China [27,45,46]. We assume that a small proportion of NCS found in the present study may be related to the sampling sites (mainly around the farmer yard), sampling time (collected in summer), and host plant species. Moreover, the heterogeneity of sampling can induce deviation in results from various field investigations.

Moreover, several studies have shown the distribution of the NCS in Asia was closely related to the certain geographical and climatic conditions. For example, *B. tabaci* Asia I was first reported in China in the 1990s, which was the most widely distributed cryptic species in Asia [47,48,49,50]. Asia II is a complex of several genetic types that have emerged more recently and have displaced the Asia I biotype in some regions [19,51]. Asia II 1 is the most widespread and damaging cryptic species in South Asia [47], particularly in India and Pakistan, where it infests cotton, tomato, tobacco, and other crops. Asia II 5 has been found in some countries, such as Bangladesh, India, Indonesia, Nepal and Pakistan [47,48,52,53]. Asia II 2 to Asia II 4, Asia II 9, Asia IV, and Asia V are less common species that are limited to China [19,46]. Asia II 6 is another complex of genetic types that has been found in China and neighboring countries [48,49,50,51]. Asia II 7 is a variant of Asia II 1 that has been detected in Pakistan, India, and Indonesia [49,52]. Asia II 8, Asia II 11, and Asia II 13 are rare cryptic species that have been reported only in India [19,52,53]. Asia II 12 is a unique cryptic species that has been identified in Indonesia [54]. Considering the specific geographical distribution of each cryptic species in Asia, it is unlikely to obtain the larger proportion of NCS through a survey conducted within a year. Thus, a long-term investigation is worth performing to illustrate the niche dynamics of native and invasive cryptic species.

Although a small proportion of NCS was also found in a previous study, the percentage of NCS (20.9%) was higher than that of MEAM1 (12.2%) in the present study, suggesting that NCS can occupy the niches colonized by MEAM1. Similarly, the NCS have not been completely displaced by the two invaders in America or Brazil, probably due to strong associations of indigenous whitefly species with specific host plants in the field [55,56]. Our study demonstrated the distribution patterns between invasive *B. tabaci* and NCS in China and their high suitability areas were not overlapping, which was consistent with the distribution of whitefly cryptic species reported in a previous study [18]. In this respect, the specific sets of high suitability niche of invasive *B. tabaci* and NCS may explain why complete replacement of indigenous species by invasive species did not occur in China. Our results on niche suitability also partly explain why invasive MEAM1 has completely replaced the indigenous A biotype from the uniform landscapes of the southern United States [57]. It should be noted that many NCS also occurred in province(s) that were not reported in the previous survey. For example, *B. tabaci* Asia II 2 was discovered in Jiangsu, Zhejiang, Anhui, and Hubei; China 1 was in Anhui, Hainan, and Henan; and Asia I was in Guangxi.

We showed that climatic and geographical locations have great influence on the distribution of *B. tabaci*. Among the ecological factors, the AMT reduced the distribution of MED but increased the distribution of MEAM1, suggesting that MEAM1 is more adaptable to high temperatures than MED. This postulate is supported by the suitable habitat analysis (Figure 3) and the previous findings that temperature-related variables had greater effects on the geographic distribution of *B. tabaci* [18]. On the other hand, AP was positively associated with the distribution of NCS but did not affect the distribution of MED (Figure 7A). The distribution of both MED and MEAM1 was negatively associated with that of NCS (Table 2), which suggests the competition replacement between invasive and native cryptic species increased in the field. These analyses demonstrate that the continued widespread distribution of NCS in China may be related to any environmental factors that may be beneficial for its competition with invasive MED or MEAM1. Indeed, accumulating evidence suggests that some ecological environmental factors are a crucial role in the continued widespread distribution of NCS in different regions of the world. For example, many NCS may have relatively high adaptability to local climatic factors, including the temperature and/or rainfall [55,56], or to local plants (i.e., weeds) and wild plants [57,58,59,60].

In the present study, three facultative endosymbionts, *Cardinium* sp., *Hamiltonella* sp., and *Rickettsia* sp., were detected in all cryptic species and the infection frequencies were significantly associated with their geographical distribution. In detail, *Cardinium* sp. was detected in a small number of MED (12.4%) and Asia II 2 (1.4%) (Figure 5; Appendix A), whereas it was not detected in any of the specimens of MEAM1. *Rickettsia* sp. infected 77.5% of MED, 39.5% of MEAM1, and 10.8% of NCS individuals. The infection frequency of *Hamiltonella* sp. was 92.3% in MED, 71.6% in MEAM1, 0.13% in Asia I, and 3.57% in China 1. The results regarding the infection frequency of endosymbionts among cryptic species were consistent with previous reports [17,61,62].

The results based on the SEM revealed relatively strong effects of climatic and geographical factors on bacterial endosymbionts, which was consistent with the previous studies [15,29]. Interestingly, AMT positively affected the infection frequencies of *Cardinium* sp. and *Rickettsia* sp. in MED, whereas it negatively affected the distribution of MED species (Figure 7). The result suggests that *Cardinium* sp. and *Rickettsia* sp. may play crucial roles in the thermotolerance of MED, although the host whitefly per se has no resistance to high temperature. Actually, *Rickettsia* sp. can influence thermotolerance in the MEAM1 [63], and the most recent research has confirmed that *Cardinium* sp. may play a crucial role in the thermotolerance of MED [64]. These research results are consistent with the distribution of *Cardinium* sp.-infected or *Rickettsia* sp.-infected MED in southern China.

In conclusion, we determined eight cryptic species from 29 geographical localities across China. Among these cryptic species, native cryptic species were still widespread in the Yangtze River Valley and eastern coastal areas. Furthermore, the climate and geographical factor have strong effects on the cryptic species distribution and their facultative endosymbionts’ infection patterns. These findings provide an insight into the role of facultative endosymbionts in the distribution and expansion of *B. tabaci* cryptic species, which will help us further understand the mechanism of successful invasion of exotic species.

## Figures and Tables

**Figure 1 insects-14-00252-f001:**
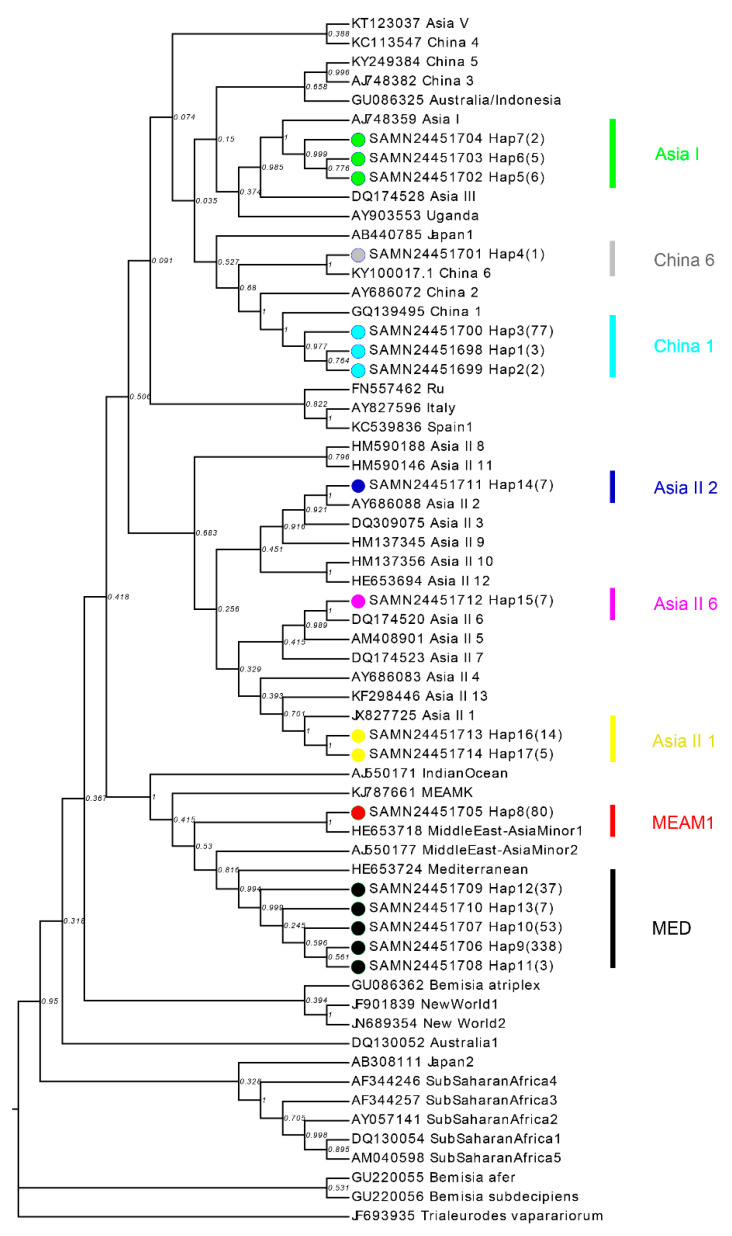
Phylogenetic tree analysis of *Bemisia tabaci* cryptic species inferred from mt*COI* sequences using neighborhood-joining method. The similar topological tree was generated using maximum likelihood method. The circles represent the sequences obtained from the current study and the other known sequences are retrieved from GenBank following the previous report by Kanakala and Ghanim, 2019. Different circle colors indicate different cryptic species of *Bemisia tabaci*.

**Figure 2 insects-14-00252-f002:**
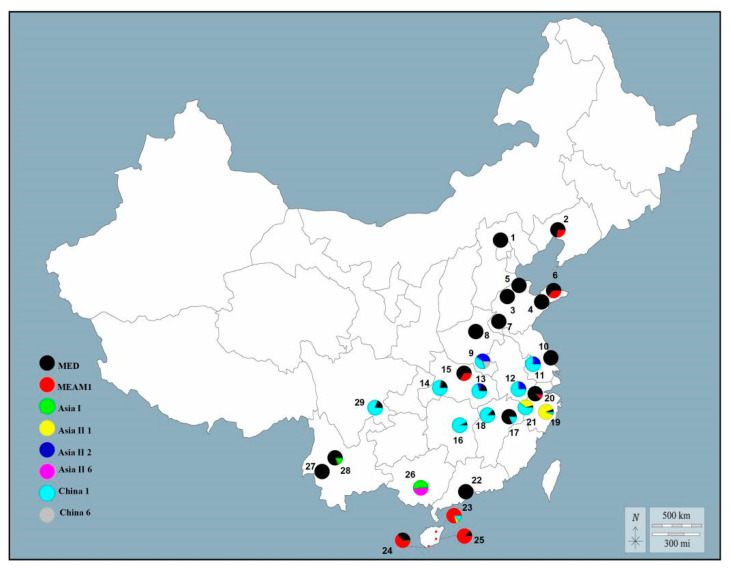
Geographical distribution of *Bemisia tabaci* cryptic species across China in 2021. The pies represent the percentage of cryptic species in each natural population; numbers indicate sampling population numbers defined in Appendix A.

**Figure 3 insects-14-00252-f003:**
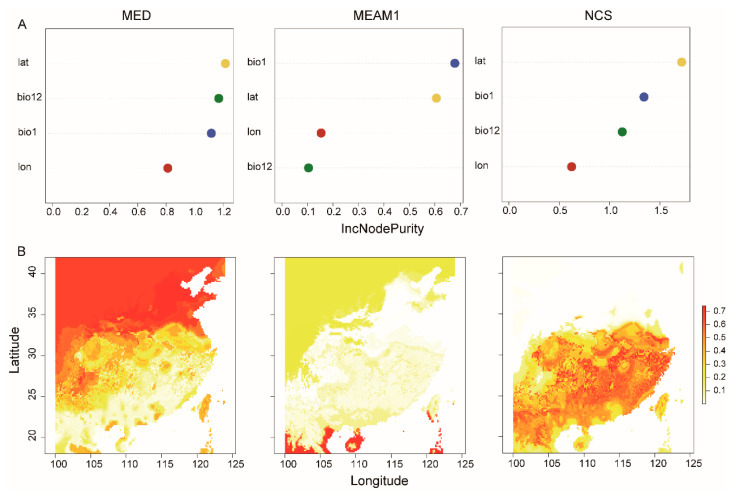
The suitable habitat distribution of *Bemisia tabaci* cryptic species in China. (**A**): the random forest model of different factors on the percentage of *Bemisia tabaci* cryptic species. Lat, latitude; Bio1, annual mean temperature; Bio12, annual precipitation; lon, longitude. The different colors represent the different environmental factors. (**B**): The suitable habitat distribution of *Bemisia tabaci* cryptic species.

**Figure 4 insects-14-00252-f004:**
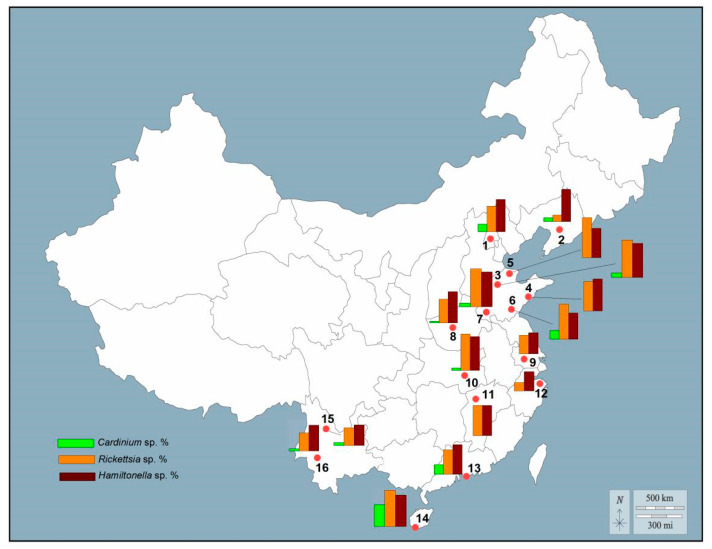
Infection frequency of three facultative endosymbionts in *Bemisia tabaci* MED populations across China. Green, yellow, and brown bars represent the infection frequencies of *Cardinium* sp., *Rickettsia* sp., and *Hamiltonella* sp., respectively. The red circles represent the sampling locations. Numbers on the map correspond to locality numbers in Appendix A.

**Figure 5 insects-14-00252-f005:**
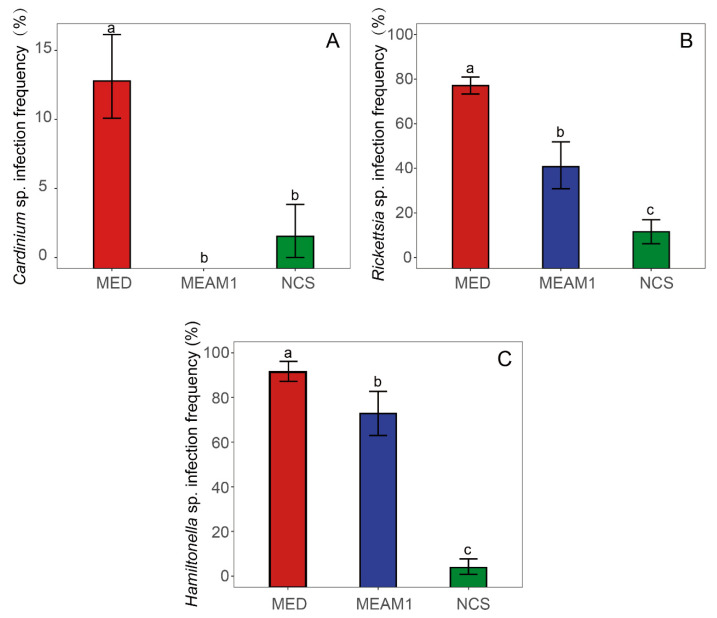
Infection frequency of *Cardinium* sp., *Rickettsia* sp., and *Hamiltonella* sp. in *Bemisia tabaci* populations. (**A**): the *Cardinium* infection frequency; (**B**): the *Rickettsia* infection frequency; (**C**): the *Hamiltonella* infection frequency. The red indicates endosymbionts’ infection frequency in the MED, blue indicates endosymbionts’ infection frequency in MEAM1, and green indicates endosymbionts’ infection frequency in NCS. Lowercase letters on the bar indicate the significant difference at the 0.05 level.

**Figure 6 insects-14-00252-f006:**
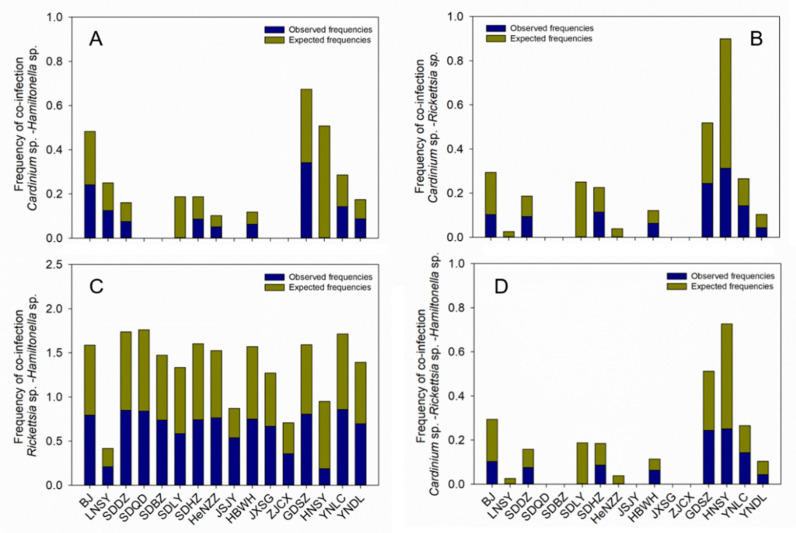
Expected frequency of coinfection based on observed frequencies in each sample. (**A**) *Cardinium* sp.–*Hamiltonella* sp., (**B**) *Cardinium* sp.–*Rickettsia* sp., (**C**) *Rickettsia* sp.–*Hamiltonella* sp., (**D**) *Cardinium* sp.–*Rickettsia* sp.–*Hamiltonella* sp.

**Figure 7 insects-14-00252-f007:**
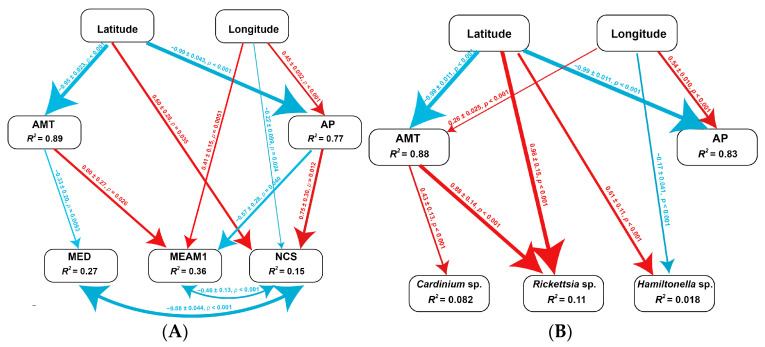
Path diagram for the structural equation model (SEM) for geographic and climatic factors and cryptic species (**A**) and facultative symbionts (**B**) in *Bemisia tabaci* populations. Statistically significant positive paths are indicated by red arrows. Statistically significant negative paths are indicated by blue arrows. The strengths of these relationships are indicated by the width of the arrows. The R^2^ values in each box indicate the amount of variation in that variable explained by the input arrows. Numbers next to arrows are unstandardized slopes. AMT, annual mean temperature; AP, annual precipitation.

**Table 1 insects-14-00252-t001:** Expected frequency of coinfection based on observed frequencies of the infections compared in each sample.

Comparison Group	Observed Value (%)	Expected Value (%)	Rank of W	*p*
*Cardinium* sp.–*Hamiltonella* sp.	7.58 ± 2.46	11.98 ± 3.53	17	0.31
*Cardinium* sp.–*Rickettsia* sp.	6.98 ± 2.40	11.33 ± 3.86	11	0.056
*Rickettsia* sp.–*Hamiltonella* sp.	64.84 ± 5.46	69.54 ± 5.34	53	0.46

**Table 2 insects-14-00252-t002:** Direct and indirect effects of environmental factors on the percentage of MED, MEAM1, and NCS estimated by the SEM model.

Effects	MED			MEAM1			NCS		
	Coefficient ± SE	z	*p*	Coefficient ± SE	z	*p*	Coefficient ± SE	z	*p*
Geographical factors	−0.26 ± 0.32	0.83	0.41	−0.21 ± 0.30	0.70	0.48	0.37 ± 0.24	1.53	0.13
Climate factors	0.61 ± 0.31	1.93	0.054	−0.22 ± 0.34	0.63	0.53	−0.47 ± 0.19	2.44	0.015
Total effects	0.34 ± 0.097	3.52	<0.001	−0.42 ± 0.18	0.18	0.019	−0.094 ± 0.16	0.60	0.55

**Table 3 insects-14-00252-t003:** Direct and indirect effects of environmental factors on the frequency of *Cardinium* sp., *Rickettsia* sp., and *Hamiltonella* sp. in *Bemisia tabaci* MED population estimated by the SEM model.

Effects	*Cardinium* sp.			*Rickettsia* sp.			*Hamiltonella* sp.		
	Coefficient ± SE	z	*p*	Coefficient ± SE	z	*p*	Coefficient ± SE	z	*p*
Geographical factors	0.17 ± 0.12	1.45	0.15	0.61 ± 0.11	5.31	<0.001	−0.062 ± 0.039	1.60	0.11
Climate factors	−0.35 ± 0.11	3.39	<0.001	−0.11 ± 0.059	1.94	0.052	−0.026 ± 0.001	0.19	0.85
Total effects	−0.19 ± 0.058	3.24	<0.001	0.49 ± 0.15	3.38	0.001	−0.088 ± 0.040	1.46	0.15

## Data Availability

All data included in this study are available, see Appendix A for details.

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
