# Peer review of "Ecological Factors Associated with the Distribution of Bemisia tabaci Cryptic Species and Their Facultative Endosymbionts"

_insects, 2023, doi:10.3390/insects14030252_

Round 1

Reviewer 1 Report (Previous Reviewer 1)

The authors have improved manuscript according to the comments and suggestions of reviewers and replied point by point to the reviewers questions. Thank you! 

Author Response

Thank you for your comments and suggestions!   

Reviewer 2 Report (Previous Reviewer 2)

I can tell that the authors put lots of effort to improve this manuscript. However, there have some errors which should addressed before it is suitable for publication.

Major suggestion:

Discussion section must need to improve.

Please, try to discuss  in one paragraph about the cryptic species distribution and abundance in other asian countries, Bangladesh, Nepal, India, Pakistan, South Korea. (Geographic distribution/ relation with major distribution of Asia II 1, Asia II 5, Asia II 7, as they are mostly distributed in Bangladesh/India, Nepal, Pakistan)

Minor:

Write all gene name in italic font. COI

Table 2 and 3: Write Geographical (G capital) and Climate (C capital)

Line 37: Write full name of mtCOI (mitochondrial cytochrome oxidase I)

Line 73: Please make it italic (Cardinium hertigii)

Line 89-90: Please write all scientific name in italic font, and check in whole manuscript. 

Line 90: Why Wolbachia spp?

Author Response

Dear reviewer:

Our manuscript ID (insects-2236499) has been revised according to the constructive comments of reviewers. Please find the revision explanation as follows:

Point 1: Please, try to discuss in one paragraph about the cryptic species distribution and abundance in other Asian countries, Bangladesh, Nepal, India, Pakistan, South Korea. (Geographic distribution/ relation with major distribution of Asia II 1, Asia II 5, Asia II 7, as they are mostly distributed in Bangladesh/India, Nepal, Pakistan).

 Response 1: Thanks for your suggestions, we have added one paragraph to make the discussion sufficiently. Please see the line 348-366 in the text.

References are as follows:

19.Kanakala, S.; Ghanim, M. Global genetic diversity and geographical distribution of Bemisia tabaci and its bacterial endosymbionts. PLoS ONE, 2019, 14(3): e0213946. https://doi.org/ 10.1371/journal.pone.0213946.

46.Jiu, M.; Hu, J.; Wang, L.J.; Dong, J.F.; Song, Y.Q.; Sun, H.Z. Cryptic species identification and composition of Bemisia tabaci (Hemiptera: Aleyrodidae) complex in Henan Province, China. Journal of Insect Science 2017, 17, 78. https://doi.org/10.1093/jisesa/iex048.

49.Lestari, S.M.; Khatun, M.F.; Acharya, R.; Sharma, S.R.; Shrestha, Y.K.; Jahan, S.M.H.; Kil, E.J.; Lee, S.; Kim, S.M.; Lee, K.Y.; Genetic diversity of cryptic species of Bemisia tabaci in Asia. Archives of insect biochemistry and physiology, 2023, 112(2), e21981. https://doi.org/10.1002/arch.21981.

50.Acharya, R.; Shrestha, Y.K.; Sharma, S.R.; Lee, K.Y. Genetic diversity and geographic distribution of Bemisia tabaci species complex in Nepal. Journal of Asia pacific entomology, 2020, 23, 509-515. https://doi.org/10.1016/j.aspen.2020.03.014.

51.Shadmany, M.; Boykin, L.M.; Muhamad, R.; Omar, D. Genetic diversity of Bemisia tabaci (Hemiptera: Aleyrodidae) species complex across Malaysia. Journal of economic entomology, 2019, 112, 75-84. https://doi.org/10.1093/JEE/TOY273.

52.Götz, M.; Winter, S. Diversity of Bemisia tabaci in Thailand and Vietnam and indications of species replacement. Journal of Asia pacific entomology, 2016, 19, 537-543. https://doi.org/10.1016/j.aspen.2016.04.017.

53.Fujiwara, A.; Maekawa, K.; Tsuchida, T. Genetic groups and endosymbiotic microbiota of the Bemisia tabaci species complex in Japanese agricultural sites. Journal of applied entomology, 2015, 139, 55-66. https://doi.org/10.1111/jen.12171.

54.Islam, W.; Lin, W.; Qasim, M.; Islam, S.U.; Ali, H.; Adnan, M.; Arif, M.; Du, Z.; Wu, Z. A nation-wide genetic survey revealed a complex population structure of Bemisia tabaci in Pakistan. Acta tropica, 2018, 183, 119-125. https://doi.org/10.1016/j.actatropica.2018.04.015

55.Ellango, R.; Singh, S.T.; Rana, V.S.; Priya, N.G.; Raina, H.; Chaubey, R. Distribution of Bemisia tabaci genetic groups in India. Environmental entomology, 2015, 44, 1258-1264. https://doi.org/10.1093/EE/NVV062.

56.Firdaus, S.; Vosman, B.; Hidayati, N.; Supena, E.D.J.; Visser, R.G.F.; van Heusden, A.W. The Bemisia tabaci species complex: additions from different parts of the world. Insect Science, 2013, 20, 723-733. https://doi.org/10.1111/1744-7917.12001.

Point 2: Write all gene name in italic font. COI

Response 2: Corrected.

 Point 3: Table 2 and 3: Write Geographical (G capital) and Climate (C capital)

 Response 3: Corrected.

 Point 4: Line 37: Write full name of mtCOI (mitochondrial cytochrome oxidase I)

 Response 4: Corrected.

Point 5: Line 73: Please make it italic (Cardinium hertigii)

Response 5: Done.

Point 6: Line 89-90: Please write all scientific name in italic font, and check in whole manuscript. 

Response 6: Done. We have checked it throughout the manuscript.

Point 7: Line 90: Why Wolbachia spp?

Response 7: Sorry for our typo, we have revised it.

Round 2

Reviewer 2 Report (Previous Reviewer 2)

Please, consider the following suggestion.

Line 348-349: Asia II 5 .............has been found in Bangladesh, India, Indonesia,  Nepal, and Pakistan.

Author Response

Dear reviewer:

  Our manuscript ID (insects-2236499) has been revised according to the constructive comments of reviewers. Please find the revision explanation as follows.

Point 1: Please, consider the following suggestion. Line 348-349: Asia II 5 .............has been found in Bangladesh, India, Indonesia,  Nepal, and Pakistan.

Response 1: Thanks for your comments, we agree with you suggestions and have revised.  Please see the line 350-351, line 355-356 in the text.

This manuscript is a resubmission of an earlier submission. The following is a list of the peer review reports and author responses from that submission.

Round 1

Reviewer 1 Report

The study 'Ecological factors associated the distribution of Bemisia tabaci cryptic species and their facultative endosymbionts' is generally well written, timely considering the climate changes and invasive species. 

Comments:

The general statements about ecology, relationships between organisms etc. should be supported by papers not only based on the data from China but also other geographical regions. 

Simple Summary: 

Line 25 Please change ..., and and negatively with Rickettsia distribution

Introduction: 

Line 51: Please add more citations, papers that also deal with other countries than China

Line 69: Please replace 'were' with 'are' and 'by' with 'on'

Materials and Methods

Line 112: what is pie marker? In the table S1 you have described 8 host plants, not 9. 

Results

Fig. 1 and Fig 2: Please add space between Asia and the Roman numeral : Asia I, Asia II, Asia III, like you did in the text

Figure 3: please replace comma after A abd B  with colon A: and B:

Figure 4: please turn the legend of colours (in the left corner) for 90 degrees, so that it is possible to read it normally. 

Figure 7: Before you used capital letters to indicate the different parts of the figure, please change here a and b to A and B

Strengths of the correlations is indicated by the width of arrows but R2=0.11 and R2= 0.83 are with similar width (in b). And also in (a), some very different R values have similar arrow widths. Please correct it. 

 Discussion

Line 315: Please change 'We here' with 'Here, we'

Line 326: you state that NCS (20.9%) was almost higher that that of MEAM1 (12.2%). What do you mean with 'almost'? Definitely, 20.9 is higher than 12.2. 

Line 344: please add the number to the citation Xue et al (2021) and add it to the reference list if it is missing

Line 373: you claim that Cardinium and Rickettsia may play crucial roles in the termotolerance of MED - actually, you did not measure termotolerance of whiteflies. You have to be cautious with bold conclusions.   

Line 381: please add 'probably could' before enhanced: ... which probably could enhance ... 

Reviewer 2 Report

This manuscript by "Ecological factors associated the distribution of Bemisia tabaci cryptic species and their facultative endosymbionts" By Li et al. evaluate genetic and geographical distribution of B. tabaci and theirs secondary endosymbionts. There are several studies on this area from different parts of world including China. However, in this manuscript, author tried to justify the scope of this study differently but many information should clarified. Please, see below for my comments and suggestions.

Line 31: Not species, correct it as cryptic species

Line 37-38: How author classified native and invasive cryptic species?

Line 40: Multiple infection was common only in MED? How about with other cryptic species?

Line 64-68: Please rephrase and summarize it.

Line 32-34: How author mention 'However, the effects of ecological factors (e.g., climatic or geographical 33 factors) on the distribution of whitefly and the infection frequencies of their endosymbionts have 34 not been fully elucidated" as they mention in line 64-68 "Although 65 endosymbionts can play an essential role in the adaptation of their host insects, the infec66 tion dynamics of facultative symbionts in the same host populations can be influenced by 67 ecological and geographic factors such as temperature, precipitation, vegetation, longi68 tude latitude and altitude [11-14].

Please see other paper "Khatun et al, Genetic diversity of Bemisia tabaci in Bangladesh and Endosymbionts profile of B. tabaci in Bangladesh, they also studied using same sample. (https://doi.org/10.1016/j.actatropica.2018.07.021

(https://doi.org/10.1007/s13199-019-00622-6)

Line 79-81: Author can cite other related papers too. 

Line 91-93: Not all the cryptic species have effect on the host reproduction, sex ratio, thermal resistance, pesticide resistance etc., Please provide the specific role, information of specific endosymbionts.

Line 104: Why author choose only these 3 endosymbionts? any specific reasons? 

Line 104-105: Is author collected whitefly only in 2021? Why not from different year or different cropping season? I think samples from only one season is not correct way to claim this kind of result and conclusion.

MM section:

Line 113: I couldn't found the sample information details in Figure 1. Please provide the sample collection details in table as main part of manuscript including Location, host plants, numbers of individuals, collection date etcs.

Line 115-118: Is author extracted DNA in PCR tubes? Please provide the DNA kit and product information details which you used for DNA extraction.

How much DNA you extracted from single individual? It was enough for identification of cryptic species and all three cryptic species from same individual? How much DNA solution was used in PCR premix for individual analysis?

I didn't found any primer informations. Please provide all the primer information with PCR conditions.

Line 126-128: Is this PCR conditions is. same for all Primers?

Line 139: Why author used Kimura-2 model? Why author construct NJ and ML phylogenetic tree? What are NJ and ML, please use full name before using abbreviations. 

In statistical analysis, how author used one way anova? Is there any replicated data? 

Line 295: Bemisia tabaci, make it Italic.

Round 2

Reviewer 2 Report

Based on this author's response, I think authors are unclear on what, how and why they did. I think they just tried to follow the previous article. As I asked them how they classified native and invasive cryptic species, they responded based on sequence alignment. It's not the correct answer which I expected.  Similarly, I asked them why they selected only three endosymbionts, and they responded as due to low frequency in MED..... This is not a logical and scientific answer and this kind of statement I didn't found inside the manuscript. How will the readers know the things in the author's mind? They should mention all the reasons clearly in the manuscript.

Why they used the Kimura-2 model?  only due to the previous paper used this method. They should be clear and need to make clear to readers. 

Author didn't indicate the revised part in the manuscript. 

I didn't find any subsequent improvement in the manuscript. Author should explain clearly in the methodology section what, how and why they did this and this.

Author Response

Dear reviewer:

Our manuscript ID (insects-2125151) has been revised according to the constructive comments of reviewers. Attached please find the revision explanation.

Sincerely,

Dong Chu

Qingdao Agricultural University, Qingdao, 266109, P. R. China

chinachudong@qau.edu.cn

################################################################################

Response to Reviewer 2 Comments

Point 1: How the authors classified native and invasive cryptic species? 

Response 1: The geographic distribution of cryptic species of Bemisia tabaci have been clarified by Kanakala and Ghanim (2019). We first determined the cryptic species of B.tabaci based on COI sequences and then classified as the native or invasive species.

The detailed process is as follows.

We firstly retrieved the reference sequences representing 43 known B. tabaci cryptic species (in-cluding native and invasive cryptic species) from NCBI database following the previous report by Kanakala and Ghanim (2019).

Then, using 17 haplotype sequences obtained in this study and 43 retrieved reference sequences, we constructed the neigh-bor-joining (NJ) and maximum likelihood (ML) phylogenetic trees in MEGA v6.0 soft-ware. Bootstrap values were based on 1,000 replicates and Bemisia afer, Bemisia atriplex, Bemisia subdecipiens, Trialeurodes vaparari-orum were used as outgroups.

Finally, the cryptic species were determined when the bootstrap value > 0.99 between haplotype sequences and retrieved reference sequences. See the line 145-151 in the text.

Point 2: Why they choose only 3 endosymbionts and why those, specifying the reason for this choice also in the text?

Response 2: All of the seven endosymbionts (Cardinium, HamiltonellaRickettsia, Wolbachia, Arsenophonus, Fritschea and Portiera) have been analyzed in B. tabaci cryptic species, only three frequent endosymbionts were exhibited in the manuscript.

The reasons are listed as follows.

Because the very smaller individuals positive-infected by Wolbachia, Arsenophonus and Fritschea in our study, the correlation analysis of Cardinium, Hamiltonella, and Rickettsia in B. tabaci MED, MEAM1, NCS were more accurate and typical. Thus, the analysis result of infection frequencies of Cardinium, Hamiltonella, and Rickettsia in B. tabaci MED, MEAM1, and NCS were exhibited in the manuscript. See the line 169-172 in the text.

Point 3: The authors have to explain why they used the Kimura-2 model? 

Response 3: 

The K2P model is the most precise model, and K2P model was used for phylogenetic analyses by Nishimaki et al. (2019).

The detailed process is as follows.

Nucleotide changes seen during the evolutionary process include substitutions, insertions, and deletions. When estimating genetic difference using the K2P model for two aligned sequences, the sites with gaps (insertions and/or deletions) are automatically removed. Thus, we selected K2P model for our phylogenetic analyses. See the line 151-157 in the text.

Point 4: The summary must be rewritten because it is unclear and fragmented and with some errors.

Response 4: We have rewritten the summary. See the line 18-29 in the text.

Point 5: Line 19 please check the spelling of "it's". I think it would be "its"

Response 5: Done.

Point 6: Line 19-22 please rearrange this sentence because is not clear

Response 6: Done.

Point 7: Line 23-24 Please check "corrected" I think it should be correlated.

Response 7: Done. We have removed it.

Point 8:  Line 104 please check the spelling of "Hamilitonella" Line 124 please check the spelling of "Hamilitonella" and check it throughout the manuscript

Response 8: Done. We have check it throughout the manuscript. 

Point 9: Line 144 please delete the space after "of"

Response 9: Done. See the line 159 in the text.

Point 10: Line 218 please check the spelling of "B. tabcai" 

Response 10: Done. See the line 238 in the text. 

Point 11: Line 257 the column MEAM1 looks purple not blue please check

Response 11: Done. See the line 278 in the text.

Point 12: I think the authors should use Cardinium spp. instead. or if I believe it is a single species Cardinium sp. The same for other bacteria.

Response 12: Thanks for your comments, we agree with you suggestions and have revised throughout the manuscript. 

Point 13: The authors should better clarify in the materials and methods paragraph how many individuals per locality used for their analyzes and why.

Response 13: At least 15 individuals per locality were sampled for whitefly species identification. After removing the B. after and B. subdecipiens individuals in some of collections, a total of 655 B. tabaci adults were prepared for the further symbiont detection and correlation analysis. The detailed description was listed in Table S1. See the line 116-118 in the text.

We are looking forward to hearing from you soon.